

# 1 An earthquake focal mechanism catalog for source and tectonic studies
# 2 in Mexico from February 1928 to July 2022.

**Quetzalcoatl Rodríguez-Pérez**[1], **F. Ramón Zúñiga**[2]
[1] Dirección de Desarrollo Científico, Consejo Nacional de Ciencia y Tecnología, Mexico City, Mexico
[2] Centro de Geociencias, Universidad Nacional Autónoma de México, Juriquilla, Querétaro, Mexico
**Correspondence:** Quetzalcoatl Rodríguez-Pérez (quetza@geociencias.unam.mx)
**Abstract.** We present a focal mechanism catalog for earthquakes that occurred in Mexico and
surrounding areas reported from February 1928 to July 2022. The magnitude of the events varies from
-0.9 to 8.2. The hypocentral depth is in the range of $0 < Z < 270$ km. Focal mechanisms in this catalog
are associated with tectonic, geothermal, and volcanic environments. Reported fault plane solutions
were derived using different types of data at local, regional, and teleseismic distances and different
methods such as first motions, composite solutions, waveform analysis, and moment tensor inversion.
So far, focal mechanism data for earthquakes in Mexico, were dispersed over many publications
without any link among them. For this reason, we collected and revised focal mechanism solutions
previously reported by different agencies and studies from published sources. Our catalog consists of
5701 fault plane solutions, and we report all the available focal mechanisms obtained by different
authors and seismological agencies for each seismic event. Additionally, we classified the fault type
into seven types: normal (N), normal – strike-slip (N-SS), strike-slip – normal (SS-N), strike-slip (SS),
strike-slip – reverse (SS-R), reverse – strike-slip (R-SS), and reverse (R) providing some preliminary
statistics results. We used the ternary diagrams of Kaverina-type classification to verify the rupture type
of focal mechanism data. We also provide a classification of the quality of the focal mechanism data



into three categories: A, B, and C. A represents good/reliable data, B represents regular, and C
represents poor/questionable data according to a well-defined criterion. Our intention is to provide a
comprehensive compilation of focal mechanism data which can help in future source and tectonic
studies in Mexico.
The earthquake focal mechanism catalog
(https://figshare.com/articles/dataset/Earthquake_focal_mechanism_catalog_for_Mexico/21663668;
Rodríguez-Pérez and Zúñiga, 2022) is given as the Supplement of this paper.
**1 Introduction**
Earthquake catalogs are used in several tasks by seismologists daily. In most cases, seismic catalogs
contain essential information such as origin time, hypocentral location, and magnitudes of the events in
a certain region. In other cases, the catalogs also include specific information such as fault planes,
source duration, seismic wave phases, seismic source parameters, and finite-fault models. Studies
related to seismicity and seismic hazard often require as input a seismic catalog that, in ideal
conditions, contains information that has been derived in a homogenous way using the same procedures
over some time. Combining different data and methods can be an alternative form to increase the
number of observations and enhance the resolution of an earthquake catalog. However, the researcher
needs to know if differences in methods or data have been incorporated to make appropriate
considerations in the analysis. This study is focused on a compilation of earthquake focal mechanisms.
Focal mechanisms describe the spatial fault orientation where earthquakes take place and the slip
direction. Fault plane solutions are essential to understanding seismotectonic processes, such as
studying the stress field in a given region. Different methods have been proposed to determine focal



mechanisms. One of the most common is based on polarities of *P*-wave motion (Knopoff and Gilbert,
1960). The moment tensor inversion was introduced later, becoming one of the most popular methods
nowadays (e.g. Dziewonski et al., 1981; Pasyanos et al., 1996; Guilhem and Dreger, 2011).
Generally, the seismic source is considered as a point source located at the hypocenter, but in other
cases, the source can be assumed as a centroid. The size of the earthquake plays an essential role in the
source representation (Dziewonski and Woodhouse, 1983). For example, the difference between the
centroid and the location of the rupture initiation can be significant. As a result, focal mechanisms
derived from wave polarities and moment tensors differ not only from inadequate velocity models or
systematic errors. Fault plane solutions obtained by wave polarities represent the geometry of the fault
at the beginning of the rupture. On the contrary, the moment tensor solutions provide the source
mechanism of the predominant component of the seismic rupture. The difference between wave
polarities and moment tensors is more drastic in the cases where the source deviates from a pure double
couple representation. Nevertheless, the seismic polarity method is still in use despite its limitations,
such as incorrect polarity readings, inaccurate velocity models, and poor azimuthal coverage of
stations, due to its simplicity and affordability. First motion polarities are often the only method to
derive focal mechanisms for small to moderate earthquakes (e.g., seismic swarms and aftershock
sequences).
At the present time, several seismological observatories routinely compute moment tensors for
earthquakes above a certain threshold of magnitude and publish their solutions in online catalogs. The
threshold magnitudes of some of these agencies are: $M_W$ = 5.0 for the Global Centroid Moment Tensor
(CMT) project (Dziewonski et al., 1981; Ekström et al., 2012), $M_W$ = 4.5 for the GEOFON Global
Seismic Network, and $M_W$ = 5.5 for the National Earthquake Information Center (NEIC) of the USGS



(Benz, 2017). Similarly, there are local and regional moment tensors catalogs with lower threshold
magnitudes ($3.5 < M_W < 4.5$). Some other online databases, such as the focal mechanism bulletin of the
International Seismological Centre (ISC) (Lentas and Harris, 2019; Lentas et al., 2019) contain both
moment tensor solutions and wave polarities of global seismicity. Focal mechanisms have been
computed and published in previous studies investigating seismotectonic features of specific regions.
Several authors have made considerable effort into determining focal mechanisms reported in different
documents and also collecting them in catalogs for specific areas to provide a set of revised
information. Many fault plane solutions are commonly spread in different documents and locations,
making standardizing information, checking, and selecting parameters a painstaking job.
In this study, we present a new catalog of focal mechanisms of earthquakes that occurred in Mexico
and surrounding areas from February 1928 to July 2022. In virtue of the relevant seismic hazard in the
region and its importance from the geodynamical perspective, many authors have computed the fault
plane solutions of seismic events using different data and several techniques (e.g., Molnar and Sykes,
1969; Dean and Drake, 1978; Chael and Stewart, 1982; LeFevre and McNally, 1985; Goff et al., 1987;
Guzmán-Speziale et al., 1989; Doser and Rodriguez, 1993; Pacheco et al., 1993; Pardo and Suárez,
1993; Pardo and Suárez, 1995; Quintanar et al., 1999; Rebollar et al., 1999; Quintanar et al., 2004;
Rodríguez-Lozoya et al., 2008; Ortega and Quintanar, 2010; Pacheco and Singh, 2010; Sumy et al.,
2013; Dougherty and Clayton, 2014; Abbott and Brudzinski, 2015; Rodríguez-Pérez and Singh, 2016;
Huesca-Pérez et al., 2022). National and International observatories also provide fault plane solutions
for seismic events generated in the territory of Mexico (e.g., the Mexican Seismological Service, SSN;
the Southern California Seismic Network, SCSN; U.S. Geological Survey, USGS; among others). This
study aims to collect and revise as many focal mechanisms as possible over time in a comprehensive
catalog.



## 2 Data and methods

### 2.1 Data

We studied earthquakes with hypocentral locations in the region corresponding to the Mexican territory and surrounding areas (latitude $12 - 33°$ N and longitude $120 - 88°$ W). Mexico is one of the most seismically active regions in the World, where different tectonic environments concur (subduction zone, transform fault zones, and intraplate regions). In Mexico, most of the seismic activity is due to the interaction among five tectonic plates (North American, Pacific, Cocos, Rivera, and Caribbean plates) and, to a lesser extent but no exempt of importance in terms of hazard, to the intraplate stresses located inland at tectonic plates. After examining information from several references in the literature and catalogs of seismological agencies, we found 5701 earthquakes with at least one fault plane solution. We reported all the available focal fault solutions obtained by different authors and seismological agencies for each seismic event making 7664 the total number of focal mechanisms. The compiled catalog has focal mechanism data from February 1928 to July 2022; the lowest data density is in the time interval of 1928-1970 (upper panel in Fig. 1). Then, the number of focal mechanisms increased gradually between 1970 to 1995 (Fig. 1). Since 1995, the number of focal mechanisms reported in Mexico has increased significantly (Fig. 1). The magnitude of these events fluctuates from -0.9 to 8.2, while the hypocentral depth is in the interval of $0 < Z < 270$ km. We classified the fault plane solutions into three categories regarding the general geological nature: 1) tectonic or regular, 2) geothermal, and 3) volcanic events.

In our catalog, tectonic earthquakes comprise 7459 focal mechanisms reported in previous studies and for different seismological observatories using different data and methods (Molnar and Sykes, 1969; Thatcher and Brune, 1971; Molnar, 1973; Johnson et al., 1976; Jimenez-Jimenez, 1977; Dean and





Drake, 1978; Ebel et al., 1978; Jimenez, 1978; Kanamori and Stewart, 1978; Yamamoto, 1978; Reyes
et al., 1979; Astiz, 1980; Morales-Matamoros, 1980; Zúñiga and Valdés-González, 1980; Chael and
Stewart, 1982; Frohlich, 1982; Natali and Sbar, 1982; Domínguez-Reyes, 1983; Havskov et al., 1983;
Astiz and Kanamori, 1984; Beroza et al., 1984; Burbach et al., 1984; González and Suárez, 1984;
González et al., 1984; Lesage, 1984; Munguía and Brune, 1984; Yamamoto et al., 1984; LeFevre. and
McNally, 1985; Singh et al., 1985; González-Ruiz, 1986; Mota-Palomino et al., 1986; Ruiz-Kitcher,
1986; Suárez and Ponce, 1986; Yamamoto, 1986; Goff et al., 1987; González-Ruiz, 1987; Yamamoto
and Mota, 1988; Yamamoto and Mitchell, 1988; Guzmán-Speziale et al., 1989; Domínguez-Rivas,
1991; Doser, 1992; Doser and Rodriguez, 1993; Pacheco et al., 1993; Pardo and Suárez, 1993; Singh
and Pardo, 1993; Wolfe et al., 1993; Zúñiga et al., 1993; Cocco et al., 1994; Ruff and Miller, 1994;
Santoyo-García-Galeano, 1994; Delgado-Vazquez, 1995; Pardo and Suárez, 1995; UNAM and
CENAPRED Seismology group, 1995; Wong et al., 1997; Pacheco and Singh, 1998; Quintanar et al.,
1999; Rebollar et al., 1999; Singh et al., 1999; Terán-Mendieta, 1999; Campos-Enriquez et al., 2000;
Cruz-Jiménez, 2000; Singh et al., 2000a,b; Delgadillo-Peralta, 2001; Rebollar et al., 2001; Iglesias et
al., 2002; Yamamoto et al., 2002; Chavacán-Ávila, 2003, Pacheco et al., 2003; Sánchez-Alvaro, 2003;
Singh et al., 2003; Zúñiga et al., 2003; Aguilar-Rosales, 2004; García et al., 2004; Núñez-Cornú et al.,
2004; Quintanar et al., 2004; Hurtado-Díaz, 2005; Bernal-Esquia, 2006; González et al., 2006;
Chavacán-Ávila, 2007; Singh et al., 2007a,b; Huesca-Pérez, 2008; Rodríguez-Lozoya et al., 2008;
Ortega and Quintanar, 2010; Pacheco and Singh, 2010; Pérez-Campos et al., 2010; Rodríguez-Lozoya
et al., 2010; Vidal et al., 2010; Jaramillo and Suárez, 2011; Martínez-López, 2011; Okal and Borrero,
2011; Stella-Ramírez, 2011; Singh et al., 2012; Soto-Peredo, 2012; Bello-Segura, 2013; Clemente-
Chavez et al., 2013; Franco et al., 2013; Rutz-López et al., 2013; Sumy et al., 2013; UNAM
Seismology Group, 2013; Yamamoto et al., 2013; De la Vega, 2014; Dougherty et al., 2014; Abbott and
Brudzinski, 2015; Singh et al., 2015; Suárez and López, 2015; UNAM Seismology Group, 2015;





Yamamoto and Jiménez, 2015; Granados-Chavarría, 2016; Gómez-Arredondo et al., 2016; Munguía et
al., 2016a,b; Rodríguez-Cardozo, 2016; Rodríguez-Pérez and Singh, 2016; Suárez et al., 2016; Vallée
and Douet, 2016; Singh et al., 2017; Yela-Portilla, 2018; Chávez-Hernández, 2019; Domínguez-Reyes
et al., 2019; Quintanar et al., 2019; Méndez-Alarcón, 2020; Singh et al., 2020a,b; Mendoza-Zúñiga,
2021; Néquiz-Guillén, 2021; Núñez-Cornú et al., 2021; Sánchez-Lopez, 2021; Corona-Fernández and
Santoyo, 2022; Huesca-Pérez et al., 2022).
On the other hand, fault plane solutions of geothermal events include 151 events reported in the
literature (Albores et al., 1980; Fabriol and Munguía, 1997; González et al., 2001; Rebollar et al., 2003;
Antayhua-Vera, 2007; Suárez-Vidal et al., 2007; Romero-Domínguez, 2013; Pérez, 2017; Oregel-
Morales, 2019; GEMex project, 2020). Finally, the volcanic earthquakes part consist of 54 focal
mechanisms (Núñez-Cornú and Sánchez-Mora, 1998; Jimenez-Jimenez, 1999; Arámbula-Mendoza,
2007; Pinzón, et al., 2017; Angulo-Carrillo, 2018; Núñez et al., 2022). Focal mechanisms reported in
this catalog were derived with the following techniques: 1) regional and teleseismic moment tensor
inversion (4747 fault plane solutions), 2) waveform analysis (208 fault plane solutions), and 3) first-
motion wave polarities of single or composite mechanisms (2584 and 125 fault plane solutions,
respectively).
**2.2 Methods**
After carefully searching fault plane solutions in the literature, we classified all the focal mechanisms
in our catalog. For this purpose, we used the FMC's computer program (Álvarez-Gómez, 2019). The
software uses the Kaverina-type classification diagrams (Kaverina et al., 1996) to verify the rupture
type of focal mechanism data. The Kaverina-type ternary diagrams classify earthquakes into seven
rupture types based on the plunges of the $P$, $B$, and $T$ principal axes: 1) normal (N), 2) normal – strike-



slip (N-SS), 3) strike-slip – normal (SS-N), 4) strike-slip (SS), 5) strike-slip – reverse (SS-R), 6)
reverse – strike-slip (R-SS), and 7) reverse (R) (lower panel in Fig. 1). Subsecuently, we calculated the
missing information of the fault/auxiliary planes, and principal axes. At this stage, we used the code
library "cmt" of seizmo software (Euler, 2014). Seizmo is a collection of different Matlab libraries to
perform different tasks in seismology. We used the function "auxplane.m" to calculate the auxiliary
focal plane. The function "sdr2tpb.m" was used to determine the principal axes of a focal mechanism.
In some cases, we had to convert the moment tensor and principal axes to strike, dip, and rake angles.
For that purpose, we used the function "tpb2sdr.m". Transformations of moment tensors to strike-dip-
rake were performed with the function "mt2sdr.m". If only the strike and dip of the fault and auxiliary
planes were reported, the rake angles were calculated with the function "GetRake" of the RFOC
software (Lees, 2018). The package RFOC is written in R language and deals with graphics for
statistics on a sphere, earthquake focal mechanisms, radiation patterns, and ternary plots.
Our database merges fault plane solutions from different studies that used different methodologies,
each with a different uncertainty level. To address this variability, we rank the quality and reliability of
the focal mechanisms in our catalog using the following criteria. We assigned a quality factor based on
data availability and the calculation process, respectively. For data availability, we consider the number
of observations, quality of the records (e.g., digitized seismograms, type of instrument), and their
spatial distribution (hypocentral distance and station coverage). In the case of the calculation process,
we consider the uniformity of the method throughout the reported study, the methodology's description,
and the method's calibration. A good calibration considers a correct selection of the medium's
properties, especially the velocity model used to calculate travel times or synthetic seismograms. Due
to the lack of uncertainty estimates reported in several studies, we do not consider them for assigning a
quality factor in most fault plane solutions. We only considered the variance reduction (VR) to assign a





quality factor when it was available. Franco et al. (2020) studied seismic moment tensors in Mexico,
and they established that a value of VR ≥ 50 % is a reasonable threshold for reliable fault plane
solutions.
We classified the focal mechanism data into three categories: A, B, and C. A represents good/reliable
data, B represents regular, and C represents poor/questionable data. Category A has one or more of the
following situations: an adequate velocity structure, a VR of > 70 %, an adequate number of
observations, a good spatial distribution of observations, a uniform methodology, a good description of
the method and data processing, and modern seismic instrumentation. The category B has one or more
of the following situations: an adequate velocity model, a VR in the range of 50 % < VR< 70 %, few
observations, a regular spatial distribution of observations, a uniform methodology, and a good
description of the method and data processing. Category C has one or more of the following situations:
a global/mean velocity model, a VR of < 50 %, few observations, poor spatial distribution of
observations, nonuniform methodology, a poor description of the method and data processing, and old
seismic instrumentation. The quality criterium presented here may help the user decide if the selected
focal mechanisms are suitable for their analysis or study. For each of the solutions, we show all the
magnitudes reported; that is, the same event can have a different type of magnitude. This is mainly due
to the difficulty of having a unified magnitude scale, since there are different types of data, and to a
greater extent, because the purpose of this study is the focal mechanisms per se.
We provide our catalog in ASCII and Excel files entitled "Focal_mechanisms_Mexico_1928-2022". In
this file, we provide the following information: 1) the number of the event, 2) the number of solutions
named as S-1, S-2, and S-*n*, where *n* is the number of a solution, 3) date of the event, 4) origin time, 5)
longitude of the epicenter, 6) latitude of the epicenter, 7) hypocentral depth, 8) manitude for each of the



solutions, 9) rupture type (N, N-SS, SS-N, SS, SS-R, R-SS, and R) , 10) strike angle 1, 11) dip angle 1,
12) rake angle 1, 13) strike angle 2, 14) dip angle 2, 15) rake angle 2, 16) plunge of the $T$-axis, 17)
azimuth of the $T$-axis, 18) plunge of the $P$-axis, 19) azimuth of the $P$-axis, 20) plunge of the $B$-axis, 21)
azimuth of the $B$-axis, 22) tectonic environment (tectonic, geothermal zone or volcanic), 23)
observations of the event (here we reported the type of magnitude for each of the solutions, $M_S$, $m_b$, $M_W$,
$M_L$, and $M_c$), 24) method used to determined the focal mechanism (first arrivals, composite solution,
waveform analysis, moment tensor), 25) variance reduction when the information was available, 26)
quality of the event, and 27) bibliographical references or seismological agency. When the information
is missing (the origin time, the seismic magnitude, or the hypocentral depth), the database cell is
highlighted in red, and a question mark is also shown in the cell.
**4 Content of the Catalogue and Discussion**
The information in this catalog is presented in an easy to understand manner as an aid to the user. The
classification of the focal mechanisms in our catalog yielded a total number of events with normal-
faulting of 1750 (Fig. 2). Earthquakes with N-SS faulting include 691 events (Fig. 3). On the other
hand, reverse-faulting forms a group of 2248 earthquakes (Fig. 4). R-SS faulting consists of 351 events
(Fig. 5). Pure strike-slip rupture is made up of 1320 seismic events (Fig. 6). SS-N faulting comprises a
group of 792 earthquakes (Fig. 7). SS-R faulting is made up of 512 seismic events (Fig. 8). In Figs. 9 to
15, we show the orientation of the pressure and tension axes. A tectonic interpretation of these data is
out of the scope of this study as it is expected to form the basis for additional future studies, but as a
basis for further analysis we provide some statistics on P and T axes for each type (Table 1) which may
serve as a guide to more detailed analysis.
Moment tensor inversion involves many assumptions and constraints that make evaluating confidence



in fault planes difficult. For this reason, we present all the solutions available for one event. In this way,
the users can consider the variability of the focal mechanisms in their analysis. The main contribution
of this work is a robust focal mechanism database for Mexico, with more than 7664 solutions for local
and regional events. The focal mechanism catalog here presented aims to be broadened and improved
to have a complete tectonic interpretation of some areas of Mexico. Fault plane solutions from this
database are intended to contribute to providing earthquake information for developing or improving
seismic hazard models in Mexico.
**5 Data availability**
Some focal mechanisms ware taken from the following sources: 1) Global Centroid Moment Tensor
(Global CMT) via https://www.globalcmt.org, 2) Mexican Global Centroid Moment Tensor via
http://132.248.6.13/cmt, 3) GEOFON Global Seismic Network via https://geofon.gfz-
potsdam.de/old/eqinfo/list.php?mode=mt, 4) International Seismic Centre (ISC) bulletin via
http://www.isc.ac.uk/iscbulletin/search/fmechanisms, 5) U.S. Geological Survey (USGS) via
https://earthquake.usgs.gov, 6) Saint Louis University moment tensor catalog via
http://www.eas.slu.edu/eqc/eqc_mt/MECH.NA, 7) SCARDEC Source Time Functions Database via
http://scardec.projects.sismo.ipgp.fr, and 8) Southern California Seismic Network (SCSN) earthquake
catalogs via http://www.eas.slu.edu/eqc/eqc_mt/MECH.NA. In all casses, last access: 17 September

19  2022.

**6 Code availability**
All figures were plotted by the Generic Mapping Tools software package (https://www.generic-
mapping-tools.org; Wessel et al., 2013). Earthquake fault classification were performed with FMC
software (https://github.com/Jose-Alvarez/FMC; Álvarez-Gómez, 2019). Conversions among fault



planes, principal axes and/or moment tensors were performed with RFOC and  seizmo cmt codes
(https://github.com/cran/RFOC; Lees, 2018; and https://github.com/g2e/seizmo).
**Author contributions.**
QRP and FRZ designed the idea and discussed the results. QRP was responsible for the data collection
and earthquake selection. The two authors contributed to the manuscript and approved the final version.
**Competing interests.**
The authors declare that they have no conflict of interest.
**Acknowledgements.**
Quetzalcoatl Rodríguez-Pérez was supported by the Mexican National Council for Science and
Technology (CONACYT) (Research - project 1126).

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





1    **Table 1.** Mean and standard deviations for the principal stress axes trends for each type of mechanism.

| Type | *T*-axis plunge | *T*-axis azimuth | *P*-axis plunge | *P*-axis azimuth |
|---|---|---|---|---|
| N | 66.81°±13.35° | 184.71°±91.23° | 19.02°±13.43° | 146.82°±106.75° |
| N-SS | 55.18°±7.35° | 190.58°±92.75° | 12.06°±7.34° | 177.73°±102.20° |
| R | 20.35°±12.27° | 184.40°±78.51° | 66.67°±12.23° | 109.95°±102.42° |
| R-SS | 14.10°±8.32° | 185.00°±104.45° | 54.38°±7.92° | 173.10°±100.26° |
| SS | 8.19°±6.07° | 216.95°±111.41° | 7.29°±5.60° | 181.58°±96.02° |
| SS-N | 30.65°±6.74° | 196.09°±108.33° | 11.82°±7.58° | 190.12°±99.39° |
| SS-R | 13.90°±8.85° | 184.44°±113.30° | 30.01°±7.37° | 172.57°±102.16° |





**Figure 1.** Magnitude time series of seismic events with at least one focal mechanism reported in this

catalog (upper panel). The Kaverina rupture type classification ternary diagram (lower panel).





**Figure 2.** Hypocentral distribution of normal faulting earthquakes (N) (upper panel). Lower panels

show magnitude and hypocentral depth distributions.

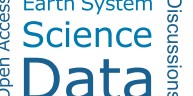

**Figure 3.** Hypocentral distribution of normal faulting with a strike-slip component earthquakes (N-SS)

(upper panel). Lower panels show magnitude and hypocentral depth distributions.





**Figure 4.** Hypocentral distribution of reverse faulting (R) (upper panel). Lower panels show magnitude and hypocentral depth distributions.



**Figure 5.** Hypocentral distribution of reverse faulting with a strike-slip component (R-SS) (upper panel). Lower panels show magnitude and hypocentral depth distributions.





**Figure 6.** Hypocentral distribution of strike-slip faulting (SS) (upper panel). Lower panels show

magnitude and hypocentral depth distributions.



**Figure 7.** Hypocentral distribution of strike-slip faulting with a normal component (SS-N) (upper

panel). Lower panels show magnitude and hypocentral depth distributions.



**Figure 8.** Hypocentral distribution of strike-slip faulting with a reverse component (SS-R) (upper panel). Lower panels show magnitude and hypocentral depth distributions.

**Figure 9.** Spatial distribution of *P*-and *T*-axes for normal faulting earthquakes (N) (lower and upper

panels, respectively).





**Figure 10.** Spatial distribution of *P*-and *T*-axes for normal faulting with a strike-slip component earthquakes (N-SS) (lower and upper panels, respectively).



**Figure 11.** Spatial distribution of *P*-and *T*-axes for reverse faulting (R) (lower and upper panels, respectively).



**Figure 12.** Spatial distribution of *P*-and *T*-axes for reverse faulting with a strike-slip component (R-SS)

(lower and upper panels, respectively).



**Figure 13.** Spatial distribution of *P*-and *T*-axes for strike-slip faulting (SS) (lower and upper panels, respectively).



**Figure 14.** Spatial distribution of *P*-and *T*-axes for strike-slip faulting with a normal component (SS-N)
(lower and upper panels, respectively).

**Figure 15.** Spatial distribution of *P*-and *T*-axes for strike-slip faulting with a reverse component (SS-R)
(lower and upper panels, respectively).