# Peer review of "An earthquake focal mechanism catalog for source and tectonic studies in Mexico from February 1928 to July 2022."

_Earth System Science Data, 2022_

## Author Comment (AC1)

**Response to comments by the reviewers and editor**

We appreciate the comments from the reviewers, which have allowed us to improve the manuscript. Overall, we have followed all the suggestions. We now give a response to the individual points raised. Changes made to the manuscript are highlighted in yellow.

**Reviewer 1.**

The authors collected and revised focal mechanisms of earthquakes which occurred inside the Mexican territory, in the period February 1928 – July 2022. For earthquakes having multiple solutions, they preferred to present all the solutions in their catalog. According to the quality of the solutions, the focal mechanisms were divided in different categories.

This catalog is a powerful tool in understanding the seismotectonic properties of Mexico, and inside the framework of the journal's aims. My recommendation is the publication of the work after minor revision.

I like the description of the methods and the data given in the introduction of the manuscript. My comments and recommendations are the following:

**Page 5 line 18. More details are needed for the magnitude distribution. -0.9 is an extreme value and if this is considered as the end member of the magnitude range does not help the reader to understand the magnitude distribution. However, it would be interesting to know the properties of the seismic network recorded so small shocks and the method used for the determination of the focal mechanism.**

We provided the information on negative magnitude events in the manuscript as requested by the reviewer.

**In Figure 1, the circles denoting the magnitudes are hidden by the vertical lines showing the frequency of the earthquakes. My recommendation is to use a different color for the circles and to plot them in front of the blue lines.**

We changed Figure 1 following the suggestions of the reviewer 1.

**Page 7 lines 13-17 What is the magnitude range of the earthquakes classified in relation to the technique used?**

We indicated in the manuscript the magnitude range for each method used to determine focal mechanisms.

**Page 9, 2nd paragraph. The authors present the criteria that used to classify the focal mechanisms in three categories. Instead of the text, a table would be very helpful for the reader to digest the classification. What does it mean adequate velocity model?**

We mean that the velocity model is specific to the region where the earthquakes are generated. Since, in many cases, average models are used that cover vast regions of the territory in Mexico.

**Page 9, lines 5,6. In the text is written: "one or more of the following situations". Is that correct or all the criteria should be fulfilled?**

That is correct. "one or more of the following situations".

**Page 9 line 24: "magnitude" instead of "manitude"**

We corrected the world.

**Page 10, line 6: "to determine" instead of "to determined"**

We corrected the verb.

**Magnitude distribution in figures 2, 3, 5, 6, 7, 8, show (more or less) two distinct groups of earthquakes. Is this used to the detectability of the existent networks, or the results of temporary networks deployed for study of the aftershock sequences?**

The reviewer correctly points out that it is a combination of these two factors. On the one side, the earthquake detection capability of permanent seismic networks has improved with new developments and densification of seismometers. On the other hand, it is also due to the use of temporary networks used to study aftershock sequences and seismic swarms. We mention this point in the manuscript.

**Reviewer 2.**

Rodríguez-Pérez and Zúñiga put together a focal mechanisms catalog, from various sources, that covers the broader region of Mexico and spans almost a century in time. They revised this dataset to maintain all the available focal mechanism solutions for each event. Further to this they used Kaverina-type diagrams to classify the focal mechanisms into different faulting types. Finally, they also assigned quality information on each focal mechanism given specific criteria. Their work aims to create a unified and homogeneous catalog that can be helpful for future studies related to tectonics and seismic hazard in Mexico.

I personally have some experience with focal mechanisms and stress inversions but in a different tectonic environment. With my comments below I hope to be able to help improve the manuscript. I think that the results of this analysis are interesting and worthy of publication but I also think that major revisions would be necessary before that. Please find below my comments which I recommend should be addressed for the manuscript to be accepted for publication.

**Major points**

**-In my opinion, the text and its structure need some work. I would recommend including a separate section for results, discussion, and conclusions. A standard IMRAD format (i.e., introduction, methods, results, and discussion) for the article is always simpler and easier to follow. Some parts of the text feel out of place and would be better placed in other sections (see comments below for this).**

We reorganized the sections of the manuscript.

**-The introduction section is brief and a bit too generic. Also, it is lacking citations (see minor points for places where I think citations could be added).**

We modified the introduction following the suggestions of reviewer 2.

**-There appear to exist a couple of earthquakes with negative magnitudes. Also, it looks like they were recorded in the 1980s. Given there are no more earthquakes with such small magnitudes looks a bit odd. Please double-check that these magnitudes are correct and update the catalog if necessary.**

We check the magnitude of these events. See response to reviewer 1.

**-The text is lacking any information on how the search for the focal mechanisms took place. For example were the individual published articles cited found via google scholar or some similar search engine? Were they found in the paper reference lists? Maybe this kind of information is implicit but I think it would be worth including the process firstly for reproducibility reasons and secondly, as this could serve as a guide for readers trying to do the same analysis in different regions.**

We provided information on this issue in the text.

**-Regarding the codes mentioned in the methods section (e.g. page 8 lines 1-12). Please make sure information about where these codes (as well as all codes used) can be found is added in the code availability section. Alternatively, another good approach would be to put all codes used in this study in a single repository (or the same where the final catalog is stored) along with 1) a very brief tutorial on how to use the codes and 2) a readme file with what the repository includes. This will ensure the reproducibility of the work. Either adding links in the code availability section or storing codes in a repository would be fine with me.**

All the codes used are mentioned in the code availability section. Each code has its repository with information on how to install and run the software with examples and documents. Putting the codes in our repository requires the permission of the authors.

**-Some parts of the text can be significantly shortened and the information they rely on can be more clearly communicated with some tables (e.g., page 9 lines 5-15; page 10 lines 13-18).**

We understand the point of reviewer 2 but keep this information as one paragraph. The information in the text is well explained.

**-Some of the criteria used when ranking the focal mechanisms in categories A, B and C are not very clear. For example "a uniform methodology" is included but I am not really sure in what sense this is meant. Also, another one is "a good description of the method", which could potentially sound a bit subjective. I would recommend that these criteria be reconsidered or at least rephrased.**

We explained these terms in the text.

**-Despite the fact that this is a dataset description paper I would still recommend adding some background context in the introduction about the broader tectonics and seismic activity (e.g., destructive large past earthquakes) in the region. This would help highlight why creating an extensive focal mechanism catalog is important.**

We added information about the background context in the introduction.

**-For the same reason as before in the discussion section, instead of stating "A tectonic interpretation of these data is out of the scope of this study" I would recommend some effort to be undertaken in discussing the results. Maybe then after the addition of some discussion, the above statement can be modified and be "A more detailed tectonic interpretation of these data is out of the scope of this study."**

We included this point in the discussion section.

**-The text is at some parts hard to follow with the tense switching from past to present. Also the terms "focal mechanism", "fault plane solutions", and "focal fault solutions" are being used on different occasions. See more detailed comments below.**

We checked the grammar of the text. We used only the term focal mechanism in the manuscript to avoid confusion.

**Minor points**

**-Page 1**

**L 21-25: This sentence is a bit hard to understand maybe something like this could help. Additionally, we classified the focal mechanisms according to their fault types using the ternary diagrams of Kaverina-type classification.**

We followed the suggestion of reviewer 2.

**L 25: We also rank the focal mechanisms into…**

The sentence was changed.

**-Page 2**
**L 2: Replace "Our intention," with "The main goal of this study is to…"**

The sentence was corrected.

**L 3: Replace which with that**

The world was replaced.

**L 13: magnitude instead of magnitudes**

The world was corrected.

**L 15: Add a citation/s at the end of the sentence**

Citations were added to the sentence.

**L 18: Add citation/s after the word time**

A citation was added to this sentence.

**L 18: Not clear what different data and methods refer to here. Please clarify.**

We clarified this point in the text.

**L 19-21: This sentence is a bit hard to follow maybe something like this could help. However, when combining different datasets it is important to...**

We rephrased this sentence.

**L 21: ...of an extensive earthquake focal mechanism catalog.**

The sentence was corrected.

**L 24: There are different methods available for determining focal**

The sentence was changed.

**-Page 3**
**L 1-2: Sentences here are hard to follow. I'd recommend something along these lines. One of the most common methods is based on P-wave polarities (Knopoff and Gilbert, 1960). Another method, that was later introduced, is the moment tensor inversion (e.g., …)**

We considered this suggestion in the version of the manuscript.

**L 5-8: These sentences here are unconnected to the flow of the text, I'd recommend deleting those.**

We decided to keep these sentences because we consider that they provide information for the introduction.

**L 8-11: Modification suggestion "Focal mechanisms derived from P-wave polarities represent the geometry of the fault at the beginning of the rupture."**

We followed this suggestion.

**L 15: that can include instead of such as**

The sentence was corrected.

**L 20: As a general practice, seismological observatories…**

We modified this sentence.

**-Page 4**
**L 8: Add a citation or more after the word information.**

Citations were added to the sentence.

**L 12: Replace "from February 1928 to July 2022" with "between 1928 and 2022".**

The sentence was corrected.

**L 14-20: Maybe break down this long reference list by mentioning the different data and/or techniques to make it easier and more useful for the user. Right now this long list of references is not helpful.**

We understand the reviewer's point, but the idea of presenting the references is so that users can find and use them and give credit to the authors of the focal mechanisms.

**L 22-24: The new paragraph could be something along these lines. "In this study, we aim to collect and revise as many focal mechanisms as possible over time in a comprehensive catalog that can be a great starting point for future seismotectonic and seismic hazard studies."**

We incorporated this suggestion to the text.

**-Page 5**

**L 5-10: This feels a bit out of place. Maybe move in the introduction.**

We moved these sentences to the introduction.

**L 12: "focal mechanisms" instead of "focal fault solutions"**

The worlds were changed.

**L 13-20: At this part of the text maybe it would be nice to include maybe the number of focal mechanisms per year or per period to make the text more specific.**

We presented  the number of focal mechanisms per period.

**-Page 6**
**see the comment made for line lines 14-20 on page 4.**

See response to  comment made for line lines 14-20 on page 4.

**-Page 7**
**L 1-6: same as the previous comment.**

See response to  comment made for line lines 14-20 on page 4.

**L 8: "focal mechanisms" instead of "fault plane solutions"**

The worlds were changed.

**L 21: What does FMC stand for? Please add the description for this abbreviation.**

The description of the abbreviation was added.

**-Page 8**
**L 5-9: Not clear the way it is written and which function was used for what analysis.**

We clarified this point in the text.

**L 9: Replace "If only" with "In cases were only"**

The sentence was corrected.

**L 15: Variability in what?**

We clarified this point in the manuscript.

**L 21: Not clear what "method's calibration" means here.**

We refer to an appropriate selection of parameters to calculate focal mechanisms or moment tensors. The sentence was corrected.

**L 24: Variance reduction (VR) is introduced here however not much information is provided about it. Some more information on this would be great.**

We introduced the meaning of VR in the text.

**-Page 9**
**L 14-15: "old seismic instrumentation" maybe "analog instrumentation" would read better.**

That is correct. We changed the world.

**L 17-19: The sentence here is unclear to me the way it is written. Given that I understood what the authors want to say my recommendation would be something along these lines. "Compiling a unified magnitude scale, given all the different magnitude scales (e.g., ML, ...,) is a demanding task that requires further detailed analysis that is outside the scope of this study."**

We rephrased this sentence.

**-Page 10**

**L 14: Remove "a total number of" and replace it with 1750. L 15: Remove "of 1750"**

The sentence was corrected.

**-Page 11**

**L 4: Not sure what "aims to be broadened and improved" means here. The way it is written, at least to me, it implies that there is more work to be done on the catalog. But ending the text on this is confusing.**

We clarified this sentence in the new version of manuscript.

**Comments on figures**

**A map view of all focal mechanisms color coded with depth and their sizes scaled according to their magnitudes is missing. Figure 1 would then be great for relying on more information about the full catalog.**

We plot focal mechanisms in a figure, but the number of focal mechanisms makes it unclear to see the details of the catalog. Figures 2-8 and 9-15 provide information about the entire catalog.

**Figure 1 (top panel). Replace the scatter plot with a density plot as after 2000 for example everything is on top of each other.**

We included a probability density function in figure 1.

**Figure 1 (lower panel). Both the text and the typical focal mechanism for each faulting type are not needed. It would be great to plot the actual focal mechanisms on this plot as small dots color coded with hypocentral depth (with transparency) to see their distribution. This will give the reader a quick idea of what kind of faulting takes place in the region.**

We modified figure 1 following the suggestions of reviewer 2.

**Figures 2-8 and 9-15 are repetitive. Not very clear why it is important to show all the different types of faulting in the main text. Maybe it is better to compile one from each of these two groups that would be included in the main text and the rest to be moved to the appendix. Replace N with number of observations. M with Magnitude.**

These figures provide information for the users of the different subsets, such as the location of epicenters, depth and magnitude distributions, and orientations of the P and T axes. We corrected the labels of figures 2 to 8.

**For Figures 9-15, the P-T axes plots, maybe it would be better to show the distribution of P (with red) and T (with blue) axes for the entire dataset (or different subsets) in a polar projection of the lower hemisphere. The size of dots can be scaled with magnitude. Or another way could be to plot rose diagrams of stress orientations.**

We modified these figures following the suggestions of reviewer 2.

---

## Author Response (AR2)

**Response to comments by the reviewers and editor**

We appreciate the comments from editors, which have allowed us to improve the manuscript. We now give a response to the individual points raised. Changes made to the manuscript are highlighted in yellow color.

**(A) Page 2, Abstract, Lines 3-5: Please replace the URL of your catalogue with the DOI https://doi.org/10.6084/M9.FIGSHARE.21663668.V1. I further suggest to modify the last sentence and remove the information on the Supplement. The data are properly published via Figshare and don't need to be declared as being a supplement to your article. I suggest to write the following:**

**"The earthquake focal mechanism catalog described in this article is available at https://doi.org/10.6084/M9.FIGSHARE.21663668.V1 (Rodríguez-Pérez and Zúñiga, 2022)"**

We rewrote this sentence following the suggestion.

**(B) Please add the same sentence (above) at the beginning of the "Data Availability Statement", followed by the description of the original data sources. Some of the sources are assigned with DOI and should be properly cited. Please write the following:**

**• "1) Global Centroid Moment Tensor (Global CMT) via https://www.globalcmt.org (Dziewonski et al, 1981; Ekström et al., 2012)"**
**the references are already included in the "References section"**

**• "4) International Seismic Centre (ISC) bulletin (https://doi.org/10.31905/D808B830, International Seismological Centre, 2022)"**
**with the following additional reference:**
**International Seismological Centre: On-line Bulletin, https://doi.org/10.31905/D808B830, 2022.**

**• "7) SCARDEC Source Time Functions Database via http://scardec.projects.sismo.ipgp.fr (Valée and Douet, 2016)"**
**with the following additional reference:**

**Vallée, M. and Douet, V.: A new database of source time functions (STFs) extracted from the SCARDEC method, Physics of the Earth and Planetary Interiors, 257, 149–157, https://doi.org/10.1016/j.pepi.2016.05.012, 2016.**

**• "8) Southern California Seismic Network (SCSN) earthquake catalogs via http://www.eas.slu.edu/eqc/eqc_mt/MECH.NA (California Institute of Technology and United States Geological Survey Pasadena, 1926)."**
**with the following additional reference:**
**California Institute of Technology and United States Geological Survey Pasadena: Southern California Seismic Network, https://doi.org/10.7914/SN/CI, 1926.**

We modified this section following the suggestions.

**(D) Please include the following reference in the "References" Section (you are citing your own data, so the full reference must be included in the "References" Section):**

**Rodriguez-Perez, Q. and Zúñiga, F. R.: Earthquake focal mechanism catalog for Mexico, https://doi.org/10.6084/M9.FIGSHARE.21663668.V1, 2022.**

We added this reference to the list.